# Multiple targeted grassland restoration interventions enhance ecosystem service multifunctionality

Shangshi Liu [1,2,3] ✉, Susan E. Ward[4], Andrew Wilby [4], Peter Manning [5], Mengyi Gong [6], Jessica Davies [4], Rebecca Killick [6], John N. Quinton [4] & Richard D. Bardgett [1,4]

The need to combat widespread degradation of grassland ecosystem services makes grassland restoration a global sustainability priority. However, simultaneously enhancing multiple ecosystem services (i.e. ecosystem service multifunctionality) is a major challenge for grassland restoration due to trade-offs among services. We use a long-term multifactor grassland restoration experiment established in 1989 on agriculturally improved, species-poor grassland in northern England, to assess how increasing the number of restoration treatments, including addition of manure, inorganic fertiliser, a seed mixture, and promotion of a nitrogen-fixing legume (*Trifolium pratense*), affects ecosystem service multifunctionality, based on 26 ecosystem service indicators measured between 2011 and 2014. We find that single interventions usually lead to trade-offs among services and thus have few positive effects on ecosystem service multifunctionality. However, ecosystem service multifunctionality increases with the number of restoration interventions, as trade-offs are reduced. Our findings highlight the significant potential for combined use of multiple targeted interventions to aid the restoration of ecosystem service multifunctionality in degraded grasslands, and potentially, other ecosystems.

Grasslands are the largest terrestrial biome on Earth[1] and support the livelihoods of more than 1 billion people[2]. Beyond their provisioning services, grasslands also provide a wide range of regulating, supporting, and cultural ecosystem services such as carbon sequestration, soil fertility maintenance, pollinator promotion, and aesthetic value[3]. However, grasslands are currently under considerable threat from multiple global change factors, including climate change, overgrazing, and excessive use of synthetic fertilisers[4,5], which are causing widespread grassland degradation and associated reductions in the delivery of key ecosystem services[6–8]. Reflecting this, the need for new

grassland restoration strategies has been identified as a global priority of sustainability policy[5,9], and is central to achieving the goals of the UN Decade on Ecosystem Restoration (2021-2030)[10].

Given that grassland degradation usually results in the loss of multiple ecosystem services[5,11], a major challenge for restoration is how best to promote ecosystem service multifunctionality (i.e. the simultaneous provision of multiple ecosystem services)[12,13]. A particular challenge here is that most restoration schemes typically focus on single interventions, which often promote a small subset of ecosystem services at the expense of others[14]. For example, the application of

[1]Department of Earth and Environmental Sciences, The University of Manchester, Manchester, UK. [2]Yale School of the Environment, Yale University, New Haven, CT, USA. [3]Yale Center for Natural Carbon Capture, Yale University, New Haven, CT, USA. [4]Lancaster Environment Centre, Lancaster University, Lancaster, UK. [5]Department of Biological Sciences, University of Bergen, Bergen, Norway. [6]School of Mathematical Sciences, Lancaster University, Lancaster, UK. ✉e-mail: liushangshi@gmail.com

**Fig. 1 | Possible mechanisms through which the number of interventions affects ecosystem service multifunctionality.** Single interventions tend to increase only a subset of ecosystem services (dashed brown arrows), and thus have limited effect on ecosystem service multifunctionality. In contrast, in ecosystems with multiple targeted interventions (green arrows), there's a higher possibility of including interventions that exert a pronounced impact on ecosystem services (here intervention B, sampling effect), and/or interventions that could lead to additive or interaction effects on individual ecosystem services (interventions C and D, additive or interaction effect), and/or different interventions that co-promote distinct subsets of ecosystem services (interventions B and C, complementarity effect).

inorganic fertiliser enhances forage production to the economic benefit of farmers, but it might negatively affect plant and soil biodiversity, and the regulating and cultural services they underpin[12,15,16]. Another issue is that single restoration interventions might not have the desired effects. For instance, while there is much evidence that increasing plant diversity promotes multiple ecosystem functions in biodiversity manipulation experiments[17–19], these benefits may not materialise in semi-naturally assembled and managed grasslands via restoration interventions that focus solely on increasing biodiversity[20–23]. Indeed, promotion of plant species richness has been shown to have a non-significant impact on most ecosystem functions in agriculturally managed grasslands[24], which indicates that restoration interventions targeted at biodiversity enhancement alone (e.g. mixed seed addition) might not be a silver bullet for promoting ecosystem service multifunctionality[25]. There is, hence, an urgent need for restoration strategies that simultaneously boost multiple ecosystem services to ensure the successful restoration of degraded grasslands.

Combining two or more interventions, rather than a single intervention, could enhance ecosystem service provision and facilitate restoration success[26,27]. Such multi-intervention restoration might involve combining multiple individual restoration strategies delivered through different pathways and targeted at different sets of ecosystem services, such as forage production, biodiversity, and carbon sequestration. For instance, mixed seed addition can promote plant diversity[28,29], with associated benefits for pollinators and cultural ecosystem services[12,30]. Low levels of fertiliser and/or manure inputs can replenish depleted soil nutrients and organic matter, thereby helping to maintain forage production and increase soil carbon sequestration[31–33], given that soil carbon accumulation is constrained by nutrient availability[34]. Although it remains untested in restoration trials, this suggests that increasing the number of restoration interventions could increase complementarity among them, thereby increasing ecosystem service multifunctionality and restoration success[14]. Alternatively, multiple interventions could lead to neutral or negative effects on ecosystem service multifunctionality, if the interventions generally exert divergent or antagonistic impacts on ecosystem services[35].

Here, we use a long-term multi-factor restoration experiment set up in 1989 on a species-poor, agriculturally improved grassland in northern England[29,36] to test whether increasing the number of restoration interventions enhances ecosystem service multifunctionality. The experiment includes all factorial combinations of four targeted grassland restoration treatments, namely the addition of farmyard manure, low amounts of inorganic fertiliser, mixed seed, and the promotion of a nitrogen fixer species (*Trifolium pratense*, red clover). This creates a gradient of number of interventions from 0 (i.e. control) to a maximum of 4, along with all possible combinations. The selected interventions are commonly used in grassland restoration schemes, and are known to individually benefit selected ecosystem services, such as forage production, soil carbon sequestration, and botanical diversity in restoration grasslands, albeit in contrasting ways[28,37–39]. We evaluate ecosystem service multifunctionality by simultaneously measuring a comprehensive range of 26 ecosystem service indicators assigned to eight groups of ecosystem services, including forage production, carbon stocks and sequestration, plant diversity conservation value, pollination service, maintenance of soil nutrients and physical stability, regulation of water quality, and aesthetic value (Table S1). Specifically, we hypothesise that: (1) single restoration interventions have limited effects on ecosystem service multifunctionality, and/or lead to trade-offs between ecosystem services; (2) increasing the number of restoration interventions enhances ecosystem service multifunctionality; and (3) multiple targeted restoration interventions mitigate the trade-offs between ecosystem services. We expect that increasing the number of restoration interventions influences ecosystem service multifunctionality by increasing the likelihood of the most influential interventions being included (Sampling effect, Fig. 1), and due to dominance of additive and synergistic effects over antagonistic effects of interventions on ecosystem services (Fig. 1). From a multifunctionality perspective, we also expect that multiple targeted interventions might also enhance complementarity between subsets of ecosystem services, thereby enhancing overall ecosystem service multifunctionality (Complementarity effect, Fig. 1). Complementarity here is defined as interventions that increase the value of a distinct subset of ecosystem services more than they diminish the value of other ecosystem services. This differs from its use in biodiversity-ecosystem functioning literatures, where complementarity refers to different species in an ecosystem synergistically enhancing the performance of each other[40].

## Results

### Grassland management and trade-offs between ecosystem services

We first asked how the identity of restoration treatments impacted individual ecosystem services. As expected, no single restoration treatment promoted all ecosystem services (Fig. 2). Moreover, the addition of both farmyard manure and inorganic fertiliser led to trade-offs between ecosystem services. Specifically, while the application of farmyard manure increased forage production, carbon stocks and sequestration, maintenance of soil fertility and aesthetic value relative to control plots (Fig. 2), it had a negative impact on soil aggregate stability (Fig. 2). Inorganic fertiliser addition alone also benefitted forage production, vegetation carbon stock, pollination services and aesthetic value, but these benefits were accompanied by a reduction in plant diversity. We also found that the addition of a seed mixture increased plant diversity and pollination, indicating its value for biodiversity conservation, but these benefits did not spill over to other ecosystem services such forage production or carbon stocks (Fig. 2). Moreover, the seeding of the nitrogen-fixing legume, red clover, increased the quality of forage production, plant diversity, flower abundance, and soil nitrogen content (Fig. 2).

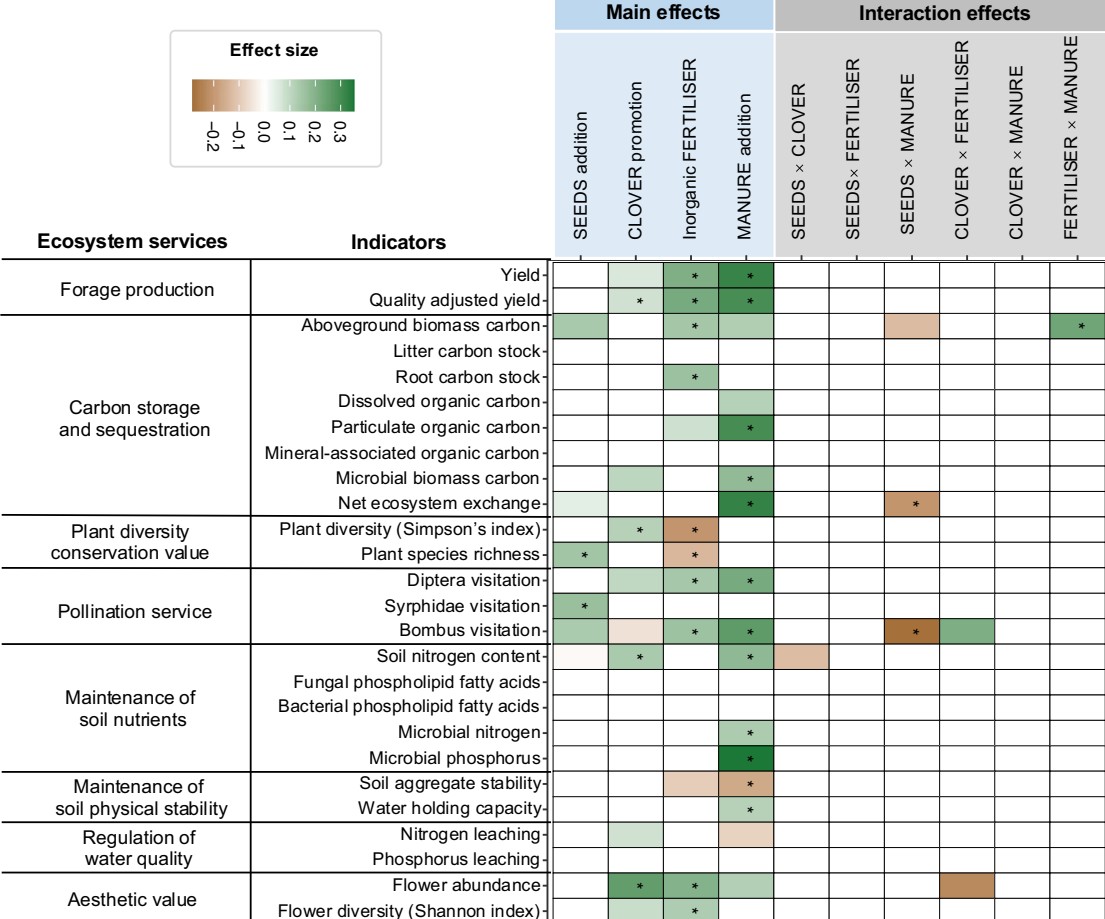

**Fig. 2 | Effects of different restoration interventions and their interactions on multiple ecosystem services.** Colours indicate the magnitude and direction of the effect of an intervention, or interaction of interventions. Effect sizes were estimated using the coefficients of linear regression models ($n = 48$ plots). * indicates statistically significant ($P < 0.05$).

## Multiple targeted interventions to promote ecosystem service multifunctionality

We next tested how the number of restoration interventions and their identity influenced ecosystem service multifunctionality. As hypothesised, we found that equally weighted ecosystem service multifunctionality increased with the number of restoration interventions (Fig. 3a and S1). Subsequent modelling showed that the models including identity of intervention had the best fit (Table S2). While individual ecosystem service indicators exhibited a generally positive, albeit variable, response to increasing number of interventions (Fig. 3b), we only detected a significant relationship for forage production, plant and soil carbon stocks, pollinator visitation, microbial phosphorus, and flower abundance (Table S3). However, when more ecosystem services were included to evaluate ecosystem service multifunctionality, they were more likely to show a positive relationship with the number of interventions (Fig. S2).

Moreover, multifunctionality indices that reflect different management objectives showed a consistently positive relationship with the number of interventions (Figs. 4 and S1). Subsequent modelling also consistently showed that the models including the identity of interventions had the best fit (Table S2), and multifunctionality generally peaked when all four restoration interventions were used (Figs. 3, 4 and S3). However, it should be noted that we observed the highest value of nature conservation-prioritized multifunctionality, which assumes the prioritization of plant diversity and pollination services, when two specific interventions, the addition of farmyard manure and mixed seeds, were used (Figs. 4 and S3).

Finally, we tested whether multiple restoration interventions could mitigate trade-offs between different ecosystem services. Our results showed that the number of interventions generally increased the evenness of multiple ecosystem services (Fig. 5a). Together with the result of ecosystem service multifunctionality calculated by the multi-threshold method (Fig. S4), these results demonstrate that multiple interventions tend to simultaneously promote the delivery of multiple ecosystem services and reduce trade-offs between them. Moreover, we found a strong positive relationship between the evenness of multiple ecosystem services and their multifunctionality, measured with equally weighted averaging (Fig. 5b).

## Discussion

Reconciling the delivery of multiple ecosystem services has been a longstanding challenge in the design of ecosystem management and restoration strategies[41–43]. Here, based on a long-term multi-factor grassland restoration experiment, we showed that while single restoration management interventions can enhance a subset of ecosystem services, they also introduce trade-offs among services and have limited effects on overall ecosystem service multifunctionality. However, we also show that increasing the number of restoration interventions increases the likelihood of reducing trade-offs and enhances ecosystem multifunctionality of grassland. As such, our results suggest that using multiple restoration interventions, rather than single interventions, is an effective tool for enhancing ecosystem service multifunctionality in grassland in the longer-term that can meet the multiple restoration goals. Overall, our findings identify a

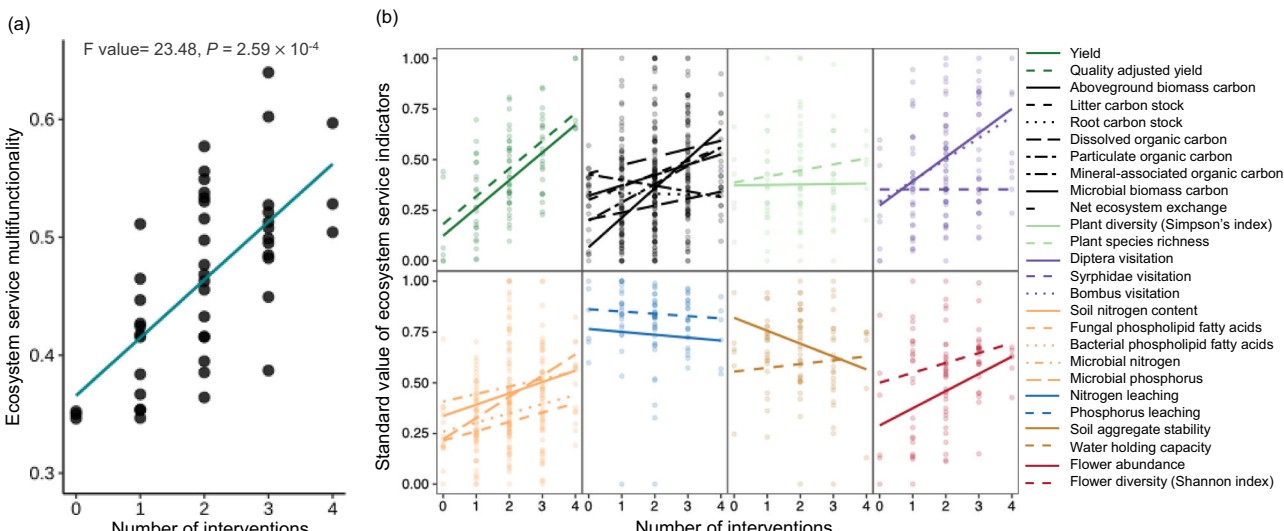

**Fig. 3 | The relationship between number of interventions and ecosystem service multifunctionality.** Ecosystem service multifunctionality was calculated by equally weighted averaging (**a**) and by assessing all individual ecosystem service indicators (**b**). Ecosystem service multifunctionality calculated using the multiple threshold method showed similar trends (Fig. S3). Statistical analyses were performed using linear mixed-effects models ($n = 48$ plots), and P values were obtained from a two-sided statistical test.

combination strategy for the restoration of degraded grasslands that could potentially be extended to other managed ecosystems to aid simultaneous delivery of multiple ecosystem services.

As expected, our results indicate that single restoration interventions have limited long-term effects on grassland ecosystem service multifunctionality, consistent with findings elsewhere[44]. For example, we found that while low amounts of inorganic fertiliser addition increased forage production and soil carbon sequestration, it decreased plant diversity, as commonly shown in past studies[45,46]. We also found that while plant diversity-oriented restoration interventions, namely the addition of a diverse seed mix, increased plant diversity, this did not translate into benefits for multiple ecosystem services. However, this lack of positive effect of plant-diversity orientated interventions on ecosystem service multifunctionality also found in other plant diversity oriented restoration projects across Europe[26,47], and may be because the relationship between biodiversity and other ecosystem services (e.g. yield and carbon sequestration) tends to saturate at higher levels of plant diversity[48], and our experiment site already had a relatively high baseline in terms of plant species richness (~23 species per 4 m² in the control plots). Moreover, the sustainable delivery of ecosystem services' multifunctionality may require the restoration of multitrophic biodiversity, rather than plant diversity alone[49,50].

In contrast to the limited effect of single restoration interventions on the ecosystem service multifunctionality, we found that grassland ecosystem service multifunctionality increased with an increasing number of restoration interventions (Fig. 3). This result aligns with common practice in restoration to combine two different approaches (e.g. fertiliser and seeding) to restore nutrient-poor degraded grasslands[51]. But we advance on this by demonstrating that further increases in the number of restoration interventions can further enhance multiple ecosystem services, as expressed by a general positive linear relationship between the number of interventions and ecosystem service multifunctionality. We propose that this is mostly attributable to different restoration interventions targeting different sets of ecosystem services in a complementary manner and balancing trade-offs between ecosystem services. This is supported by the positive relationship between the number of interventions and the evenness of ecosystem services. The evenness of multiple ecosystem services also positively associated with their multifunctionality (Fig. 5).

This implies that one of the underlying mechanisms of multiple interventions is that they generally balance trade-offs between ecosystem services and enhance multiple ecosystem services more evenly. This relationship could also be attributed to an additive or synergistic impact of restoration treatments on the same ecosystem services. For example, the addition of inorganic fertiliser and farmyard manure synergistically increased above-ground biomass. While we also observed antagonistic effects (i.e. of manure addition and seed addition on pollinator visits and net ecosystem exchange), such effects are less common than additive and synergistic effects in our study.

While the number of restoration interventions can be a predictor of ecosystem service multifunctionality, it is important to note that the identity of the intervention selected will also determine the outcomes of grassland management (Table S2, Fig. S3). Optimising multiple targeted interventions requires careful selection of actions, because different combinations of interventions can have varying consequences for the outcome of grassland management. This is because, first, the benefits of multiple interventions lie in their complementarity, rather than in their similarity of impact. In other words, restoration interventions should target different facets of ecosystem structure and functioning to enhance ecosystem complexity[52]. Moreover, additive effects or synergistic interactions would make combinations of interventions more effective than individual interventions. For example, the combined use of soil disturbance and seeding can additively or synergistically interact to enhance specific ecosystem service indicators (e.g. native species richness and biomass)[26,51,53]. Therefore, restoration interventions can be optimised if synergies between them are encouraged and antagonistic interactions are minimized.

We also emphasise that the best combination of specific intervention strategies will depend on ecological context, such as the type of grassland. It is essential to recognize that grasslands are not a homogenous ecosystem type; they range from species-rich, semi-natural habitats with high conservation value to intensively managed, agriculturally improved grasslands that meet the demand for food production. These differences have profound implications for the ecosystem services they provide and the most appropriate management and restoration interventions required. For instance, while fertiliser addition may enhance productivity in agriculturally improved grasslands, such practices should be avoided when restoring

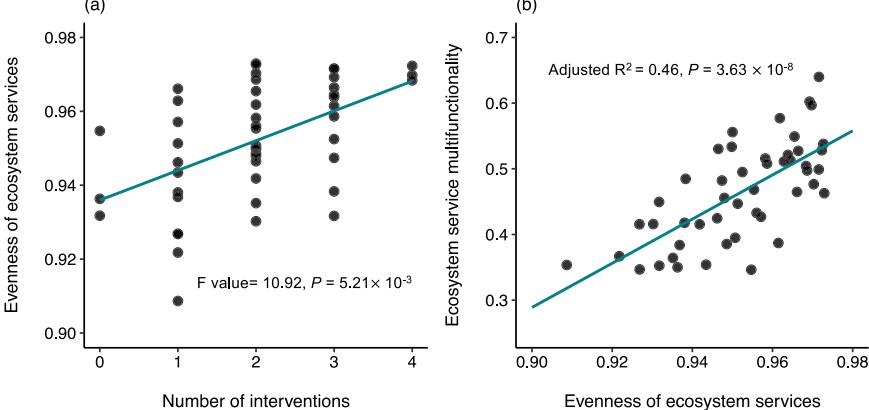

**Fig. 4 | The relationship between number of interventions and ecosystem service multifunctionality indices representing scenarios of different management objectives.** Ecosystem service multifunctionality was measured by: (**a**) regenerative agriculture prioritized; (**b**) nature conservation prioritized; (**c**) climate mitigation prioritized; and (**d**) aesthetic value prioritized averaging of standardised values of different ecosystem services. These ecosystem service multifunctionality indices assigned high proportional weights to forage production, plant biodiversity and pollination, carbon stocks and sequestration, and aesthetic value, respectively, as shown on the y-axis of (**a**–**d**). Statistical analyses were performed using linear mixed-effects models ($n = 48$ plots), and $P$-values were obtained from a two-sided statistical test.

**Fig. 5 | The relationship between number of interventions and trade-offs between different ecosystem services.** The relationship between the number of interventions and evenness of standardised score of multiple ecosystem services (**a**), and the relationship between evenness of multiple ecosystem services and ecosystem service multifunctionality (**b**). Ecosystem service multifunctionality was calculated by equally weighted averaging. Statistical analyses were performed using a linear mixed-effects model for (**a**) and simple linear regression for (**b**), respectively, $n = 48$ plots. $P$-values were obtained from a two-sided statistical test.

grasslands for biodiversity. Therefore, restoration approaches must be carefully considered and tailored to the specific ecological context and restoration objectives of each grassland type. Lastly, the selection of restoration interventions should be guided by the management objectives. Determining the exact nature of the best interventions to achieve ecosystem service multifunctionality would require more detailed information on stakeholder priorities, for example, as assessed in social surveys[54], as well as the financial and labour costs of implementing these interventions.

Our work provides an important step toward understanding the role of multiple interventions in promoting ecosystem service multifunctionality. Here, we focused on four widely used restoration interventions for agriculturally improved temperate grasslands[55,56], future studies will be needed to evaluate the potential benefits of further increases in the number of interventions beyond these four interventions. For example, adding optimized grazing pressure, diversified livestock, or soil inoculation may further enhance ecosystem service multifunctionality[49,57–59]. Given that ecosystem restoration is occurring at far larger scales than those investigated here, future studies should also explore how combinations of different restoration methods impact ecosystem service multifunctionality across diverse locations and at large spatial scales, to inform the design of multiple targeted interventions at the landscape scales[5,60].

Future studies would also benefit from wider observation and quantification of ecosystem service multifunctionality. Although we measured a comprehensive range of functions that underpin crucial ecosystem services, our measurements were not exhaustive. For example, beyond carbon stocks and sequestration, a full assessment of climate change mitigation service should also consider nitrous oxide and methane flux to provide a full picture of greenhouse gas balance[61]. The wider environmental footprint (e.g. antibiotic resistance) of restoration interventions also needs to be considered[62,63]. Current methodology for measuring pollination services would also be improved by taking measurements at multiple time points and larger spatial scales to enhance representativeness. Moreover, from a temporal perspective, the stability and sustainability of ecosystem service multifunctionality should also be explored in future studies via long-term observations[64]. This is particularly important given global warming and the increasing frequency of extreme climate events[65,66], which could destabilize the ecosystem service multifunctionality of grasslands[67,68] and other ecosystems[69].

Overall, our study provides a robust 'proof of concept' for a combined approach to grassland restoration, laying the groundwork for testing multiple interventions to sustain the delivery of ecosystem services across a wider range of sites and ecosystems. We acknowledge that an increasing number of interventions may inevitably require more financial investment for restoration, which may limit the scalability in resource-constrained regions. Nevertheless, the interventions used here are all commonly used in grassland management, albeit usually singularly, and do not involve large additional investment in resources or machinery. A growing number of environmental land management schemes offer financial support for farmers to restore degraded grasslands, and these could be effectively employed to fund grassland restoration schemes using multiple interventions, as proposed here[5,70–72]. Nevertheless, results from our long-term restoration experiment emphasize the need to move beyond the use of single interventions as a solution to restore degraded grasslands, and potentially other degraded ecosystems, and to balance competing management goals and reverse the negative impact of global environmental changes on ecosystem service multifunctionality.

## Methods
### Multiple factorial grassland restoration experiment
Our study was based on a long-term grassland management experiment at Colt Park Meadows, Ingleborough National Nature Reserve (Latitude 54°12'N, Longitude 2°21'W, 350 m.a.s.l.) in northern England, United Kingdom[36,73–75]. The experiment was established in 1989 and provides a range of management-relevant experimental restoration treatments at a field scale[36]. The soil is a shallow brown-earth of moderate-high residual fertility over limestone, and the plant community, when the experiment was established, was described as an agriculturally improved, plant species-poor grassland dominated by *Lolium perenne* and *Cynosurus cristatus*. These agriculturally improved grasslands are widespread across northern Europe and are the target of various Environmental Land Management Schemes aimed at enhancing biodiversity and ecosystem service multifunctionality[76]. This restoration experiment included four treatments with their respective controls, namely: low amount of inorganic fertiliser addition (nitrogen: phosphorus: potassium = 20:10:10) at 25 kg nitrogen ha$^{-1}$ y$^{-1}$; farmyard manure addition at 12 t ha$^{-1}$ y$^{-1}$; addition of both commercial and locally sourced mixed seeds of 19 species to increase plant diversity; and, the promotion of the nitrogen-fixing leguminous herb *Trifolium pratense* (red clover) by seed addition, which has been shown to increase soil nitrogen and soil carbon sequestration[38]. The inorganic fertiliser and mixed seed addition treatments were established in 1990, the farmyard manure addition treatment in 1998, and the clover seed addition treatment was added in 2004 and repeated in 2011. Combined in a fully factorial design, this gave 16 different restoration treatment plots, each repeated in 3 blocks in a split plot design to give a total of 48 sampling plots (each 3 × 3 m). This design provides a gradient of the number of restoration interventions, ranging from 0 (i.e. control) to a maximum of 4 interventions, along with all possible combinations, allowing for the exploration of the effect of the number of interventions on ecosystem service multifunctionality.

### Measurement of individual ecosystem services
Measurements were made over a period of 4 years (2011–2014) to investigate the effects of the 4 restoration treatments, and all their combinations, on a comprehensive range of ecosystem service indicators, including forage production (i.e. hay yield and quality-adjusted hay yield), carbon stocks and sequestration (i.e. aboveground and root biomass, litter and soil carbon stock, microbial carbon content, and net ecosystem exchange), plant diversity conservation value (i.e. plant species richness and Simpson index), pollination (i.e. the abundance of the main taxa of pollinators, namely bumblebees, hoverflies, and other flies), the maintenance of soil nutrients (i.e. soil nitrogen content, microbial nitrogen and phosphorus content, fungal and bacterial biomass) and soil physical stability (i.e. soil aggregate stability and water holding capacity), the regulation of water quality (i.e. leaching from soil of nitrogen and phosphorus), and aesthetic value (i.e. flower abundance and diversity). For those ecosystem service indicators that are greatly influenced by the difference in biotic and abiotic conditions between seasons and years, such as forage production, net ecosystem exchange, and flower abundance and diversity, we conducted multiple measurements in the field.

### Forage production
Forage production was measured annually by sampling aboveground shoot tissue from a 2 × 1 m area within each treatment plot in July or August from 2011 to 2014. Vegetation samples from each plot were analysed for total yield (dry weight of vegetation) and for carbon and nitrogen content. Crude protein content was estimated as 6.25 × Nitrogen content, and quality-adjusted yield was quantified as biomass yield × crude protein content[77].

### Carbon stocks and sequestration
Carbon stocks were evaluated based on the measurement of multiple plant and soil carbon pools. In addition to aboveground biomass, we measured root biomass and litter carbon stock for each plot in

summer 2013. All plant samples were dried, weighed, and grinded before measuring the carbon content. Bulk soils were sampled in summer 2013 to a depth of 10 cm, and bulk density was determined from cores of known volume. Given that different soil carbon fractions have distinct formation pathways, persistence, and functioning, and thus play different role in the delivery of carbon sequestration service, we separated soil organic carbon from bulk soil into different soil organic matter fractions by the wet sieve[78,79]. This fractionation method resulted in three fractions: (1) particle organic matter fraction, which is mainly formed from plant residues and has high chemical recalcitrance; (2) mineral-associated organic matter fraction stabilised through chemical bonding to minerals; and (3) dissolved organic carbon fractions[80,81]. Briefly, fresh soil (10 g dry soil equivalent) samples were size fractionated to particle organic matter (>53 $\mu m$), mineral-associated organic matter (0.45-53 $\mu m$) and dissolved organic matter (<0.45$\mu m$). Dried soil fractions were ground in a ball mill and were analysed for total carbon and nitrogen as above. Total carbon of plant and soil samples was measured by combustion and gas chromatography (Elementar Vario EL CN analyser). Given that there was no detectable inorganic carbon in the soil samples, total carbon is taken to equal organic carbon. Soil organic carbon stock per unit area (g C m$^2$) was calculated from carbon concentrations (% C) and bulk density measures (g cm$^{-3}$). Moreover, we measured microbial biomass carbon using the chloroform fumigation-extraction procedure[82].

Carbon sequestration was indicated by net ecosystem exchange of $CO_2$, which was measured from July 2011 to August 2014 over the midday period, at monthly intervals during the growing season (May–Sept), and bi-monthly from Oct–April. Measurements of $CO_2$ exchange were made over 120 s intervals using an EGM4 portable infrared gas analyser (PP systems, USA) coupled to a customised chamber lid (30 cm diameter and 35 cm height) fitted to a base ring sunk to 5 cm soil depth[83].

### Plant diversity conservation value
A vascular plant species survey was carried out in each plot in 2014, the cover of each species was recorded from the central 2 × 2 m area, and plant richness and Simpson index were used to indicate the conservation value of plant diversity.

### Pollination service
Pollination service was indicated by the visitation of pollinators[84]. Observations of insect visitations to flowering plants were made in the June of 2014 when grazing animals were excluded from plots. Observations were carried out between 10:00 am and 3:00 pm where possible on warm sunny days (12 °C or higher <30% cloud cover or >15 °C when overcast) without rain or strong winds (>10 mph). The observation quadrat was 2 m × 2 m, with the quadrat placed at the centre of the treatment plot. All observations were for 10 min. Bees, hoverflies and other true fly visits to flowers were recorded. Flower visitors were identified to species level for bees and genus level for hoverflies, though here we report data for the major taxonomic groups: Bumblebees (*Bombus* spp. comprising 98.5% of bee visits), hoverflies (Diptera: Syrphidae), and other flies (Diptera excl. Syrphidae). Together these groups composed 99.2% of all flower visits recorded.

### Maintenance of soil nutrients
The maintenance of soil nutrients was indicated by soil total nitrogen content, soil microbial biomass element content and phospholipid fatty acid (PLFA) content[85]. Soil microbial biomass carbon, nitrogen and phosphorus were measured using the chloroform fumigation-extraction procedure[82]. A further sub-sample of the 2013 soils was analysed for soil microbial communities by PLFA, using the extraction, fractionation and quantification procedure to obtain soil fungi and bacterial biomass[82].

### Regulation of water quality
Regulation of water quality was indicated by the capacity of soils to retain nutrients. The capacity of soils to retain nutrients was measured in the laboratory in 2013, using intact soil cores from the field site, subjected to a drying and re-wetting treatment[86]. Soil cores (6 cm diameter, 10 cm depth) with vegetation were collected from random areas within each field plot and transferred to a temperature and humidity-controlled cabinet, set at 20 °C with 16 h daylight. The dry-re-wetting treatment allowed soil cores to dry to 20% water holding capacity, then re-wetted to 60% water holding capacity. After the re-wetting, cores were leached with 150 ml distilled water, passed through the soil core 3 times. The resultant leachate was analysed for nitrogen and phosphorus concentration, and the volume leached noted.

### Maintenance of soil physical stability
The maintenance of soil physical stability was indicated by water holding capacity and soil aggregation. Water holding capacity was measured by the water content held in the intact soil cores after excess water had drained away. Soil aggregate stability was measured by fast wetting approach[87]. Soil samples were rapidly immersed in water for 10 min and then transferred through a $50\mu m$ sieve that was already immersed with ethanol to separate fractions. Mean weighted diameter was calculated to indicate soil aggregate stability.

### Aesthetic value
The aesthetic value was indicated by the abundance and diversity of flowers[60,88,89]. Observations were conducted eight times during the peak flowering period (mid-June to mid-July) of 2014 to quantify flower abundance and diversity. A 1 m × 1 m quadrat was positioned at the centre of the plot, and the number of open flowers for each species within the quadrat was recorded. Flower abundance was expressed as the average number of flowers per square metre over eight observations, while the Shannon index was used to determine flower diversity based on the abundance of flowers of each flowering plant species[88].

### Evaluation of ecosystem service multifunctionality
We calculated ecosystem service multifunctionality using two commonly used approaches: the averaging method and the multi-threshold method[13,44,90]. The averaging method takes the average standardised value of multiple services to form a single index of ecosystem performance. First, temporal repeat measurements were averaged prior to subsequent calculation. We then standardised each ecosystem service indicator to a range of 0-1, where variables which are negatively related to ecosystem service provision (i.e. the nutrient leaching) were subtracted from 1 so that positive values relate to improved service provision. We employed an equal-weighted averaging of standardised score of all 26 ecosystem services indicators to calculate ecosystem service multifunctionality. Moreover, to quantify the trade-off between ecosystem services, we also calculated the evenness of ecosystem service indicators using the 'vegan' package in R[91], by treating 26 ecosystem service indicators as species and standardised score of ecosystem service indicators as abundance[92].

We also calculated four multifunctionality indices that aim to represent the capacity of the ecosystem to meet diverse management objectives[93]. These multifunctionality indices weighted the ecosystem services according to scenarios of weighting representing four possible management objectives: regenerative agriculture, nature conservation, climate mitigation, and aesthetic value, which assigned high weights to forage production, plant biodiversity and pollination, carbon stocks and sequestration, and aesthetic value, respectively. These scenarios were designed to reflect the likely management goals and ecosystem service priorities of several major stakeholder groups in the study region, such as the agriculture sector, environmentalists, policy makers, and tourists, despite evidence

that all stakeholders demand multiple ecosystem services[54,89]. We first calculated standardised scores for each of the 8 ecosystem service groups by averaging standardised score of their respective indicators. Then, we applied different weightings for ecosystem service groups. The prioritized ecosystem service group received a 50% weighting, while other groups that support the prioritized service collectively received the remaining 50% weighting. For instance, in the case of regenerative agriculture multifunctionality scenario, forage production was weighted at 50%. Carbon stocks and sequestration, pollination services, maintenance of soil fertility, and maintenance of soil physical stability each received a 12.5% weighting. For detailed information on the weighting of different ecosystem services in our scenarios, please refer to Figure S6.

Although the averaging method provides an intuitive way to quantify multifunctionality, it should be noted that this metric assumes substitutability between ecosystem services[94]. To address this issue, we also calculated ecosystem service multifunctionality using a multi-threshold method. In this method, ecosystem service multifunctionality was evaluated by the number of services that simultaneously exceed multiple thresholds. In this study, results that come from using this method are similar to those measured using the averaging method (Fig. S4), thus, we only present those based on the averaging method in the main text.

### Statistical analysis

To explore potential redundancy and trade-off among ecosystem services, we first conducted principal component analysis (PCA) on the whole ecosystem service indicator dataset (Fig. S5).

To test the first hypothesis, we estimated the effect of the type of the restoration interventions on individual ecosystem service indicators using linear mixed-effects models, where the intervention effect and/or the interaction of the intervention were modelled as the fixed effects component and the block effects were treated as random component. Split plot design was specified as a nested random effect but was excluded in the results due to singular fits. Where necessary, response variables were log-transformed to improve linearity. Hypothesis tests for the effect of the treatment or their interaction were carried out by stepwise selection based on the Akaike information criterion (AIC). In particular, the effect of the treatments was tested first, followed by the effect of the interactions. We checked for model mis-specification through a range of standard residual plots. To compare the effects of restoration interventions on multiple ecosystem service indicators, we calculated standardised effect sizes of the interventions on ecosystem service indicators generated from the mixed models. Due to the statistical and ecological difficulty of fitting and interpreting high-order interactions between treatments (Table S4), we did not include three- and four-way interactions in the model[95,96].

To test the second hypothesis that increased numbers of restoration treatments lead to increased ecosystem service multifunctionality, we again used the linear mixed-effects modelling approach. The baseline model has the number of restoration interventions as a continuous fixed effect component, the block effect and the type of intervention combinations as random components. The effect of the number of restoration interventions on ecosystem service multifunctionality could be a result of: (a) plots with multiple interventions having a higher probability of including a dominant intervention with a strong effect on multifunctionality; or (b) different interventions having an additive or interaction effect on multifunctionality. Therefore, if a significant effect of the number of restoration interventions was detected, we continued to investigate the model with the random effect from the type of intervention combinations replaced by the presence of the interventions (intervention identities model), and their pairwise interactions (intervention interactions model). The proportions of

random variation explained by the interventions and/or their interactions provide insight into the contribution of specific intervention combinations to the improvement of ecosystem service multifunctionality.

To answer the question of whether multiple interventions simultaneously increase different ecosystem services, in addition to the multifunctionality data calculated by the multi-threshold method, we tested the effect of the number of interventions on the evenness of individual ecosystem service indicators using a mixed effect model. We used the same testing approach as the one used in testing the second hypothesis. Moreover, simple linear regression was used to test the impact of evenness of ecosystem service indicators on ecosystem service multifunctionality.

All analyses were done in R[91], using the 'lme4' package[97] for mixed effect modelling and visualised with the 'ggplot2' package[98].

### Reporting summary

Further information on research design is available in the Nature Portfolio Reporting Summary linked to this article.

## Data availability

All ecosystem service multifunctionality data generated in this study have been deposited in the *Figshare* (https://doi.org/10.6084/m9.figshare.28440020.v1).

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

## Acknowledgements

This research was initiated and led by RDB with grant support from Defra (project number BD5003) and NERC (project number NE/T012307/1). We thank the Natural England team from Colt Park meadows, especially Colin Newlands and Andrew Hindes, for help and providing access to the site. We also thank our colleagues Helen Quirk, Catherine Baxendale, Mike Whitfield, Gareth McShane, and Phil Donkersley for assistance in the collection and processing of samples; Tanya St Pierre for pollinator observation data; and Judith Allinson for vegetation survey data. Finally, we are grateful to Richard Brand-Hardy and Oliver Edmunds of Defra, and to Chris Chesterton and Steve Peel of Natural England, who provided support and guidance during early stages of the project, and to Roger Smith who had the foresight to establish the Colt Park experiment in 1989.

## Author contributions

S.L.: Conceptualization; formal analysis; writing-original draft, review and editing. S.E.W.: Data curation; writing-review and editing. A.W.: Data curation; writing-review and editing. P.M.: Conceptualization; writing-review and editing. M.G.: Formal analysis; writing-review and editing. J.D.: Writing-review and editing. R.K.: Formal analysis; writing-review and editing. J.N.Q.: Writing-review and editing. R.D.B: Conceptualization; funding acquisition; research supervision; writing-review and editing.

## Competing interests

The authors declare no competing interests
