## [Transparent Peer Review file · Nature Communications]

Multiple targeted grassland restoration interventions enhance ecosystem service multifunctionality

Corresponding Author: Dr Shangshi Liu

Version 0:

Reviewer comments:

Reviewer #1

(Remarks to the Author)

This paper reports on a factorial study in which four different grassland restoration interventions were applied individually and simultaneously to assess their effects on 26 different ecosystem functions. The strength of the study is the many different functions the authors measured 10-15 years after the experiment was implemented. They report a number of interesting results about tradeoffs between interventions and how multiple interventions tend to increase multifunctionality. The paper is mostly clearly written.

That said, my main critique of this paper is the broad sweeping statements that are made about restoration interventions based on an experiment in which treatments were conducted in 3 x 3 m plots replicated 3 times each at a single site. Having conducted restoration experiments in grasslands and other ecosystems at multiple sites, I have always found that some results are consistent across sites, but many vary depending on site. To make credible recommendations to practitioners, restoration methods need to be tested at multiple sites. Moreover, the UN Decade on Ecosystem Restoration is aiming to restore millions of hectares globally and increasingly restoration projects are on the order of tens to hundreds to thousands of hectares. Even small grassland restoration projects are typically on the order of a few hectares, making it questionable how much one can predict from 3 x 3 m plot studies for restoration. For example, nutrient cycling effects may be swamped by variation in soil. Pollinators forage quite differently depending on variation in the surrounding vegetation, which will be quite different in a 3 x 3 m compared to a much larger intervention plot. The authors say in the introduction that multiple interventions have not been tested in “a real world restoration setting” (line 84) which is largely true, but this experiment is not a “real world restoration setting”. This isn’t to say that there is no value from this study, but the authors need to tone down their broad sweeping claims about the application to restoration considerably. It feels like this study was set up as a BEF study that is being applied after the fact to restoration.

The authors make the case (line 58) that often restoration interventions are applied individually rather in the combined manner. They also say that they chose their four interventions – farmyard manure, inorganic fertilizer, seeding, and planting an N-fixing species – because they are widely used in this grassland ecosystem. Are these interventions typically applied individually in this ecosystem? I have rarely seen restoration practitioners plant or seed in nutrient poor soils without some form of fertilizer, whether that be fertilizer, N-fixing species, or manure (the latter typically at smaller scales).

Detailed comments

In the abstract and main text, the authors says that “26 ecosystem service indicators were measured over 4 years”. This sentence is misleading, as it suggests that all 26 ecosystem service indicators were measured in 4 years. In fact, most of the ecosystem functions were measured in a single year. So. it would be much more accurate to say that they were measured in 1-4 years.

It would also be more accurate to say that the measurements were taken 12-15 years after the initiation of the experiment. Since the experiment was started in 1989, I assumed that this was at least a 20-yr experiment, but the last data were collected in 2014, a decade ago.

Line 60. I found it odd that the authors used “ceasing to use fertilizer” as the example here. Reducing nutrient inputs is frequently discussed in high fertility grasslands to reduce invasive competition, but in this study three of the “restoration” treatments are some sort of manipulation to increase nutrient availability.

Line 74. It seems like it should be “two or more” interventions. I can’t imagine anybody would specifically suggest two. And could cut “It has been proposed”; that is implied with the citation of references.

Figure 2. It is interesting that there are so few interaction effects, but this may be due to low power given that each treatment combination had an n of 3. Did the authors do any power analyses?

Figure 1. Why is the green arrow from the brown circle to “high multifunctionality” thicker than the others? I don’t think panel B adds much to the figure as it is subsumed by information in A.

Line – 139 – sentence could be deleted and just start with “as expected, no single”

Lines 162-163. It says that the identity of the intervention was significant in the model of ecosystem multifunctionality, but I didn’t see it clearly stated which intervention or two resulted in the highest multifunctionality. Likewise, this result is discussed generally but not with respect to specific interventions on Lines 257-258.

On line 170 the authors say that their multifunctionality indices reflect different stakeholder priorities. I assumed based on that sentence that they had stakeholders what their priorities were. In fact, the authors subjectively placed the different functions into categories. So, the reference to stakeholders is misleading.

Figure 5 – panel B. Aren’t the calculation for “ecosystem service functionality” and “evenness of ecosystem services” based on the same numbers? It seems like these two measures aren’t independent of one another, though admittedly, I wasn’t entirely clear how each of these calculations was done.

Lines 276 – This paragraph points out some limitations to the study but doesn’t note the replication or scaling issues discussed above.

Lines 284-289 – This is a long, jargon-filled sentence that says there is a need for different targeted interventions to inform multiple targeted interventions, which doesn’t need to be stated.

More generally, there is a fair amount of repetition of generalities in the discussion. The text could be tightened up.

Line 400. It sounds like pollination services were measured observing individual plots of 2 x 2 m for 10 minutes in one month. This seems like a highly insufficient sample size to generalize about pollination services. And as noted previously, I question whether pollination services (at least as applied to restoration) can be assessed meaningfully in such a small plot as the measurements in one plot are necessarily going to be influenced by adjacent plots.

Line 447 – Should be “species”. Specie is gold coin.

Line 497 – “split” is misspelled.

Figure S3. The tick label size needs to be increased to be legible.

Reviewer #2

(Remarks to the Author)

The authors present data on how multiple management interventions affect multiple ecosystem services in grassland ecosystems. This is an important and timely topic, and the authors did a good job of measuring and interpreting important services. I thought the measurements of pollinator visitation and aesthetic measures of flower abundance and diversity were especially novel.

The analysis was appropriate for the objectives of the study. The work is explained in detail and could be reproduced by others. I did have a few suggestions on revising the paper.

Line 381. Are there any carbonates in your soil and how were they dealt with? What was the pH of the soil? Carbonates can be large components of some grassland soils and they have to be analyzed as well as SOM.

I thought the usage of evenness measures in the ES part of the paper was somewhat novel and interesting. Can you cite some of the evenness literature in this section?

I agree with the authors that we need more consideration of grazing intensity in these kinds of studies. It was nice to see that they acknowledged that in the discussion.

Reviewer #3

(Remarks to the Author)

Multiple targeted grassland restoration interventions enhance ecosystem service multifunctionality
Liu et al.

We* are glad to have had the opportunity to review the manuscript "Multiple targeted grassland restoration interventions enhance ecosystem service multifunctionality" by Liu et al.

This work covers an essential and critical topic, especially in light of global climate change, biodiversity crises, and the

ongoing demand for increased agricultural production while conserving nature.

The manuscript focuses on the restoration of agriculturally improved grasslands, examining how different restoration interventions impact ecosystem service provision and ecosystem multifunctionality. The novel and valuable aspect of the study arises from the thorough sampling and from the opportunity to observe the long-term impacts of restoration interventions using multiple treatments. The comprehensive data gathering and analysis cover a wide range of ecosystem services and their combined multifunctionality, offering an integrated and thorough knowledge of the ecosystem improvements achieved.

The fact that the study focuses on grasslands is also relevant - grasslands are often overlooked in conservation and restoration efforts. Despite their critical role in ecosystem service provision and biodiversity, grasslands are experiencing significant degradation worldwide, including in Europe, as also clearly pointed out by the authors.

The manuscript is interesting, well-composed, covers an important and timely topic, and has high applicable value for real-life restoration actions. The methodology and analysis are sound and well explained, meeting the quality requirements and standards of the field. However, few aspects can be further elaborated or clarified.

General comments:

The study focuses on improved agricultural grasslands, while the introduction and discussion cover grasslands in a very general sense. It should be clear from the introduction to the discussion that the study considers (and obtained results apply to) only one type of grasslands - species poor agriculturally improved grasslands. It is important, as grasslands *sensu lato* include a variety of ecosystems, ranging from intensively managed agricultural grasslands to semi-natural and natural habitats with native, characteristic, self-assembled plant compositions. The range of potential services they CAN provide varies greatly (e.g. Lindborg et al. 2023), irrespective of the stakeholder interests. Thus, the restoration goals also vary, together with the set of suitable restoration interventions. For example (as also shown in this study), fertilization and seed addition can decrease species richness and homogenize species composition. In ancient (never plowed, never seeded, or fertilized) semi-natural grasslands that hold very high conservation value (e.g. Heinsoo et al. 2020), it would be almost a crime to promote prioritization of biomass, vegetation carbon stock, or even perhaps overall ecosystem multifunctionality if it threatens those conservation values. Thus, to avoid inadvertently promoting practices that could be harmful in some cases, please add a short paragraph acknowledging that there are multiple grassland types, each with different values and uses, and each requiring its own set of restoration interventions. It must be clear that careful consideration of approaches is needed depending on the target ecosystem.

It might also be that improved service provision or increased multifunctionality at the observed time point does not always mean a 'better' or more resilient system in the long run, which should also be addressed. See also DeCock et al., 2023.

The discussion could be revised to reduce some repetitiveness of the findings and add a more thorough synthesis with previously published research on ecosystem multifunctionality. If we may recommend, our own recent study had quite a similar approach but showed that both multitrophic diversity and ecosystem service multifunctionality could be simultaneously increased with grassland restoration (Prangel et al. 2024).

Very small details that might be helpful to authors:

Figure 3 (b) is quite hard to read; it is not easy to understand which group is in which figure (especially as the legend and figures are not in the same order, at least in the end - soil related vs nitrogen-phosphorus seems to be switched). Consider changing the legend or adding subfigure titles.

Lines 45-46: Suggest removing "also" from the sentence and adding "provisioning services" to the list. "Grasslands also provide a wide range of provisioning, regulating, supporting, and cultural ecosystem services such as food production, carbon sequestration..."

Lines 79-83: This sentence is quite long and contains a lot of information. It could be split into two sentences.

Line 84: change 'word' to 'world'.

Lines 87-89: "Alternatively, multiple interventions could lead to neutral or negative effects on ecosystem service multifunctionality if the interventions generally exert divergent or antagonistic impacts on ecosystem services." Please add an example.

Line 215: Add 'than' to "As such, our results suggest that the use of multiple restoration interventions, rather than single interventions..."

Lines 295-296: Add "be" in the sentence. "The wider environmental footprint (e.g., antibiotic resistance) of restoration interventions also needs to be considered."

Line 298: make a correction: "...multifunctionality should also be explored in future studies via long-term observations."

Lines 405-406: make a correction: "The observation quadrat was 2 m × 2 m, with the quadrat placed at the center of the treatment plot."

Line 440: "flower" should be in plural. "The aesthetic value was indicated by the abundance and diversity of flowers."

Line 447: make a correction: "...flowering plant species."

Line 459: make a correction: "we also calculated the evenness of ecosystem service indicators using the 'vegan' package..."

Line 530: make a correction: "Moreover, simple linear regression..."

References

DeCock, E., Moeneclaey, I., Schelfhout, S., Vanhellemont, M., De Schrijver, A., Baeten, L., 2023. Ecosystem multifunctionality lowers as grasslands under restoration approach their target habitat type. *Restor. Ecol.* 31, 1–10.

<https://doi.org/10.1111/rec.13664>

Heinsoo, K., Sammul, M., Kukk, T., Kull, T., & Melts, I. (2020). The long-term recovery of a moderately fertilised semi-natural grassland. *Agriculture, Ecosystems & Environment*, 289, 106744.

Lindborg, R., Hartel, T., Helm, A., Prangel, E., Reitalu, T. and Ripoll-Bosch, R., 2023. Ecosystem services provided by semi-natural and intensified grasslands: Synergies, trade-offs and linkages to plant traits and functional richness. *Applied Vegetation Science*, 26(2), p.e12729.

Prangel, E., Reitalu, T., Neuenkamp, L., Kasari-Toussaint, L., Karise, R., Tiitsaar, A., Soon, V., Kupper, T., Meriste, M., Ingerpuu, N. and Helm, A., 2024. Restoration of semi-natural grasslands boosts biodiversity and re-creates hotspots for ecosystem services. *Agriculture, Ecosystems & Environment*, 374, p.109139.

*This review was compiled jointly by an early-career researcher and the advisor.

Elisabeth Prangel, Aveliina Helm (University of Tartu, Estonia)

Reviewer #4

(Remarks to the Author)

"I co-reviewed this manuscript with one of the reviewers who provided the listed reports. This is part of the Nature Communications initiative to facilitate training in peer review and to provide appropriate recognition for Early Career Researchers who co-review manuscripts."

Version 1:

Reviewer comments:

Reviewer #1

(Remarks to the Author)

I appreciate that the authors have addressed many of my and other reviewers' comments by rewording portions of the text which has improved the paper. I particularly recognize the effort they made by adding the power analyses.

However, a few of my initial concerns remain. First, the authors still make multiple broad sweeping statements that are made about restoration interventions based on an experiment in which treatments were conducted in 3 x 3 m plots replicated 3 times each at a single site. I list a few examples of their overstatements below...

Line 83, 101-102 – the authors say that multiple interventions have not been tested in "real world restoration trials". Their 3 x 3 m plots are not "real world restoration trials" though they imply that in lines 101-102. They are reporting on small-scale restoration experiments. As I mentioned in the last review, they over-sell the scale and real-world practicality of their results. Also, the text is nearly identical in two adjacent paragraphs, so it is redundant. I strongly recommend deleting the sentence in line 100-102.

Line 220 says that the authors "Overall, our findings identify a novel strategy for the restoration of grasslands that could potentially be extended to other managed ecosystems to aid simultaneous deliver of multiple ecosystem services." Again, this is inaccurate. They don't identify a novel restoration strategy. These are all standard restoration strategies that are sometimes implemented at the same time and sometimes not – e.g. lots of managers seed, including N fixing species, and inorganic fertilizer in grasslands and other terrestrial ecosystems. If seen that combination used in temperate grasslands and forests by restoration practitioners across multiple hectares and not just in experimental plots. Most practitioners don't use both inorganic fertilizer and manure because of the costs of multiple interventions.

Line 262 says "While the number of restoration interventions can be a strong predictor of ecosystem service multifunctionality". Figure 3b does not suggest that this relationship is "strong". Yes – some of the lines increase with the number of interventions, but many of them are flat and a couple go down. I think it should just "can be a predictor." Figure 1b. I still don't think panel B adds much to the figure as the same information is in panel A in a different format. It seemed clear from figure A and the multiple times the authors state it in the text that they think the number of interventions will increase ecosystem multifunctionality.

On line 170 the authors say that their multifunctionality indices reflect different scenarios stakeholder priorities. In the figure they call them "different management objectives". I think the latter word is more accurate and should be used throughout in the text which would simplify the text considerably.

Lines 287-295. I appreciate that the authors have now explained that stakeholders surveys would be necessary to assess stakeholder priorities. However, I think the paragraph could be shortened considerably by just saying that "The selection of restoration interventions should be guided by management priorities." The rest of the text is fairly long-winded without saying much. I think that throughout (e.g. line 486) the authors should switch from stakeholder priorities to management objectives.

In the closing paragraph the authors only passingly consider the costs of implementing multiple restoration interventions, but costs are a key consideration for land managers. Yes these methods are commonly used but managers tend to pick and choose the ones that they find to be most effective to achieve their goals given cost limitations.

More generally, there is a fair amount of repetition of generalities in the discussion. The text could be tightened up, despite

the fact that the authors say they already did that.

Pollinator measurements. I still contend that observing individual plots of 2 x 2 m for 10 minutes in one month is insufficient to characterize pollinators. I know the authors measured many variables but that doesn't mean that how they measure different variables should be of low rigor.

Reviewer #2

(Remarks to the Author)

I thought that the authors did a great job in responding to the reviewers and the revision is greatly improved.

Reviewer #4

(Remarks to the Author)

Responses to reviewer comments in blue

REVIEWER COMMENTS

Reviewer #1 (Remarks to the Author):

This paper reports on a factorial study in which four different grassland restoration interventions were applied individually and simultaneously to assess their effects on 26 different ecosystem functions. The strength of the study is the many different functions the authors measured 10-15 years after the experiment was implemented. They report a number of interesting results about tradeoffs between interventions and how multiple interventions tend to increase multifunctionality. The paper is mostly clearly written.

Response: Thank you for your constructive comments and for highlighting the strength of our study.

That said, my main critique of this paper is the broad sweeping statements that are made about restoration interventions based on an experiment in which treatments were conducted in 3 x 3 m plots replicated 3 times each at a single site. Having conducted restoration experiments in grasslands and other ecosystems at multiple sites, I have always found that some results are consistent across sites, but many vary depending on site. To make credible recommendations to practitioners, restoration methods need to be tested at multiple sites. Moreover, the UN Decade on Ecosystem Restoration is aiming to restore millions of hectares globally and increasingly restoration projects are on the order of tens to hundreds to thousands of hectares. Even small grassland restoration projects are typically on the order of a few hectares, making it questionable how much one can predict from 3 x 3 m plot studies for restoration. For example, nutrient cycling effects may be swamped by variation in soil. Pollinators forage quite differently depending on variation in the surrounding vegetation, which will be quite different in a 3 x 3 m compared to a much larger intervention plot. The authors say in the introduction that multiple interventions have not been tested in “a real world restoration setting” (line 84) which is largely true, but this experiment is not a “real world restoration setting”. This isn’t to say that there is no value from this study, but the authors need to tone down their broad sweeping claims about the application to restoration considerably. It feels like this study was set up as a BEF study that is being applied after the fact to restoration.

Response: We appreciate your critique regarding the replication and spatial scale of our restoration experiment, and have toned down our conclusions on the application of our findings to restoration as suggested. Please see our specific responses and revisions below.

We certainly agree that the outcome of restoration may vary across different sites, and that experiments conducted across multiple locations are crucial for making broad

recommendations to practitioners. However, we view our study, which is uniquely based on a long-term grassland restoration experiment and comprehensive range of ecosystem measurements, as an initial step towards this goal. Specifically, our findings provide important 'proof of principle' for the largely untested notion that a combinational approach to restoration interventions is the best way to achieve decisive progress in ecological restoration (Rillig et al. 2024). Moreover, while we acknowledge that our study was not conducted at a large-scale (e.g., ten to thousands of hectares), the use of the Colt Park experiment does provide a unique system for experimentally testing the efficacy of a combinational approach to grassland restoration interventions, using multiple and far-ranging measures of ecosystem service indicators that would be logistically challenging at larger scales. For example, measuring net ecosystem exchange alone required repeat (bi)monthly field work over four years, which would be very challenging to achieve on a large scale across many sites. However, to recognize the important points made by the reviewer we now explicitly state in the discussion of the revised manuscript that further research across diverse locations is needed to generalize our findings

“Given that ecosystem restoration is occurring at far larger scales than those investigated here, future studies should also explore how combinations of different restoration methods impact ecosystem service multifunctionality across diverse locations and at large spatial scales, to inform the design of multiple targeted interventions at the landscape scale^{5,55}.” (page 14, lines 303-306).

“Current methodology for measuring pollination services would also be improved by taking measurements at multiple time points and larger spatial scales to enhance representativeness.” (page 15, lines 314-316).

“Overall, our study provides a robust ‘proof of concept’ for a novel combined approach to grassland restoration, laying the groundwork for testing multiple interventions to sustain the delivery of ecosystem services across a wider range of sites and ecosystems.” (page 15, lines 322-324).

Regarding the comment about "real-world restoration setting" wording, we used this to highlight the gap in recent theoretical research (e.g., Rillig et al., 2024) and field applications intersect. Moreover, the Colt Park experiment was originally set up to investigate the long-term restoration of grassland diversity in agriculturally improved grassland using common widely used interventions, rather than to test diversity-functioning relationships as suggested by the reviewer (Smith et al. 1996). However, we recognize that our experiment does not encompass the full complexity of large-scale interventions, and have revised the "real-world restoration setting" to "real-world restoration trials" to tone down accordingly (Page 4, Lines 82 and 102), and that hopefully our results will inform real-world practice, as was highlighted as a strength of our study by Reviewer #3 and #4.

Overall, we appreciate the suggestion of the reviewer to tone down broad claims, and revised the manuscript accordingly to ensure our conclusions are presented with appropriate caution and clearly state the limitations of our study in the discussion section. Our aim is to avoid overgeneralization and provide balanced recommendations that reflect the scope of our research.

References

Rillig, M. C., Lehmann, A., Rongstock, R., Li, H., & Harris, J. (2024). Moving restoration ecology forward with combinatorial approaches. *Global Change Biology*, 30(6), e17361.

Smith, R. S., Corkhil, P., Shiel, R. S., & Millward, D. (1996). The conservation management of mesotrophic (meadow) grassland in Northern England. 2. Effects of grazing, cutting date, fertilizer and seed application on the vegetation of an agriculturally improved sward. *Grass and Forage Science*, 51(3), 292-305.

The authors make the case (line 58) that often restoration interventions are applied individually rather in the combined manner. They also say that they chose their four interventions – farmyard manure, inorganic fertilizer, seeding, and planting an N-fixing species – because they are widely used in this grassland ecosystem. Are these interventions typically applied individually in this ecosystem? I have rarely seen restoration practitioners plant or seed in nutrient poor soils without some form of fertilizer, whether that be fertilizer, N-fixing species, or manure (the latter typically at smaller scales).

Response: We agree that restoration practitioners often combine fertiliser and seeding in ecosystems with nutrient-poor soils. However, it is rare for these interventions to be applied factorially and over long timeframes in controlled studies, and the majority of the restoration experiment literature focuses on single restoration interventions (Nolan et al., 2021; Slodowicz et al., 2023; Rillig et al., 2024). For instance, the use of seed mixes is widely recommended as a crucial intervention for the recreation of species-rich meadows without fertiliser addition (Freitag et al., 2012). However, to address the reviewers point, we now acknowledge that some restoration practices already contain several interventions, though also that the individual and cumulative impact of these are not often evaluated in a rigorous fashion, as done here.

“This result aligns with common practice in restoration to combine two different approaches (e.g., fertiliser and seeding) to restore nutrient-poor degraded grasslands⁵¹. But we advance on this by demonstrating that further increases in the number of restoration interventions can further enhance multiple ecosystem services, as expressed by a general positive linear relationship between the number of interventions and ecosystem service multifunctionality.” (page 12, lines 243-244)

Reference:

Freitag, M., Klaus, V. H., Bolliger, R., Hamer, U., Kleinebecker, T., Prati, D., ... & Hölzel, N. (2021). Restoration of plant diversity in permanent grassland by seeding: Assessing the limiting factors along land-use gradients. *Journal of Applied Ecology*, 58(8), 1681-1692.

Nolan, M., Dewees, S., & Ma Lucero, S. (2021). Identifying effective restoration approaches to maximize plant establishment in California grasslands through a meta-analysis. *Restoration Ecology*, 29(4), e13370.

Rillig, M. C., Lehmann, A., Rongstock, R., Li, H., & Harris, J. (2024). Moving restoration ecology forward with combinatorial approaches. *Global Change Biology*, 30(6), e17361.

Slodowicz, D., Durbecq, A., Ladouceur, E., Eschen, R., Humbert, J. Y., & Arlettaz, R. (2023). The relative effectiveness of different grassland restoration methods: A systematic literature search and meta-analysis. *Ecological Solutions and Evidence*, 4(2), e12221.

Detailed comments

In the abstract and main text, the authors says that “26 ecosystem service indicators were measured over 4 years”. This sentence is misleading, as it suggests that all 26 ecosystem service indicators were measured in 4 years. In fact, most of the ecosystem functions were measured in a single year. So. it would be much more accurate to say that they were measured in 1-4 years.

It would also be more accurate to say that the measurements were taken 12-15 years after the initiation of the experiment. Since the experiment was started in 1989, I assumed that this was at least a 20-yr experiment, but the last data were collected in 2014, a decade ago.

Response: We agree that stating "26 ecosystem service indicators were measured over 4 years" may give the impression that all indicators were measured annually over the four-year period. This was not our intention: for some ecosystem service indicators that show temporal variation (e.g., net ecosystem exchange and yield), we conducted up to four years of measurement, but for others (e.g., soil carbon storage) we conducted one-off measurement. Regarding the duration of this experiment, the experiment started in 1989, and the first and last data were collected in 2011 and 2014, which was 22-25 years after the start of this experiment.

We have revised the text following the suggestion to more accurately reflect our methodology.

“Using a long-term multifactor grassland restoration experiment established in 1989 on agriculturally improved, species-poor grassland, we assessed how increasing the number of restoration treatments, including addition of manure, inorganic fertiliser, a seed mixture, and promotion of a nitrogen-fixing legume (Trifolium pratense), affects

ecosystem service multifunctionality, based on 26 ecosystem service indicators measured between 2011-2014.” (page 2, lines 25-31)

Line 60. I found it odd that the authors used “ceasing to use fertilizer” as the example here. Reducing nutrient inputs is frequently discussed in high fertility grasslands to reduce invasive competition, but in this study three of the “restoration” treatments are some sort of manipulation to increase nutrient availability.

Response: We now have rephrased this sentence following the suggestion to better align the example with the focus of this study.

“For example, the application of inorganic fertiliser enhances forage production to the economic benefit of farmers, but it might negatively affect plant and soil biodiversity, and the regulating and cultural services they underpin^{12,15,16.}”(page 3, lines 58-60)

Line 74. It seems like it should be “two or more” interventions. I can’t imagine anybody would specifically suggest two. And could cut “It has been proposed”; that is implied with the citation of references.

Response: We have changed this sentence as suggested (page 3, lines 73-74)

“Combining two or more interventions, rather than a single intervention, could enhance ecosystem service provision and facilitate restoration success”

Figure 2. It is interesting that there are so few interaction effects, but this may be due to low power given that each treatment combination had an n of 3. Did the authors do any power analyses?

Response: Yes, we detected very few interaction effects, indicating that the effects of restoration interventions on ecosystem service indicators are mostly additive. Unfortunately, no power analysis was conducted to determine the number of replication when the experiment was established in 1989. We totally agree that a larger sample size would better detect interaction effects, which is why we estimate the interactions using mixed effect model, rather than ANOVA, to mitigate this issue to some extent. With appropriate model selection using mixed effect models, it is possible to identify the interactions important to the changes in the response variables. Since the models used all 48 samples, the number of observations contributing to the estimation of the treatment effect is larger than the number of observations per treatment combination (n=3). For example, when we estimate the interaction between two treatments (e.g., inorganic fertiliser and seeds addition), the model can estimate the interaction effects from all plots that received these treatments and compare them to their counterparts (e.g., inorganic fertiliser + seeds addition vs. control; inorganic fertiliser + seeds addition + manure addition vs. manure addition). In other words, we have up to 24

observations when we assess the treatment identity effect and up to 12 observations when we assess the two-way interaction effect. Therefore, the detectable effect size in practice will be smaller than the effect size required for $n=3$ observations. We now include a power analysis of treatments on ecosystem services in the supplementary files (Table S4). In general, our current experimental design has enough sample size to detect a 20% change in multifunctionality indices, and a 40% change in most ecosystem service indicators, with a statistical power of 0.8.

Table S4. Required sample sizes for different effect sizes (percentage change to the mean), given a statistical power of 0.8. These calculations are based on the means and variances of the ecosystem service indicators and multifunctionality indices. The relationship between effect size and sample size, with a statistical power of 0.8 for all ecosystem service indicators and multifunctionality indices, was illustrated using a t-test (Harrer et al., 2021). The analysis was conducted using the "pwr" package (Champely, 2020) in R. Required sample sizes smaller than 24 are in bold.

	Change 20%	Change 30%	Change 40%	Change 50%
Yield	53	24	14	10
Quality adjusted yield	71	32	19	13
Aboveground biomass carbon	70	32	19	12
Litter carbon stock	69	32	18	12
Root carbon stock	51	24	14	9
Particulate organic carbon	68	31	18	12
Mineral associate organic carbon	33	15	9	7
Dissolved organic carbon	101	46	26	17
Microbial biomass	50	23	14	9
Net ecosystem change	30	14	9	6
Plant diversity (Simpson's index)	56	25	15	10
Plant species richness	40	19	11	8
Diptera visitation	54	25	15	10
Syrphidae visitation	69	31	18	12
Bombus visitation	149	67	38	25
Soil nitrogen content	24	12	7	5
Microbial nitrogen	26	12	8	6
Microbial phosphorous	61	28	16	11
Fungal PLFA	69	32	18	12
Bacterial PLFA	54	25	15	10
Soil aggregation stability	15	8	5	4
Water holding capacity	18	9	6	4
Nitrogen leaching	10	5	4	3
Phosphorus leaching	7	4	3	3
Flower abundance	42	19	12	8
Flower diversity (Shannon index)	21	10	6	5
Equal weighted averaging ecosystem service multifunctionality	5	3	3	2

Regenerative agriculture prioritized ecosystem service multifunctionality	13	7	5	4
Nature conservation prioritized ecosystem service multifunctionality	6	4	3	3
Climate mitigation prioritized ecosystem service multifunctionality	6	4	3	3
Aesthetic value prioritized ecosystem service multifunctionality	8	5	3	3

Reference:

Champely, S. (2020). pwr: Basic Functions for Power Analysis. R package version 1.3-0, <https://CRAN.R-project.org/package=pwr>.

Harrer, M., Cuijpers, P., Furukawa, T.A., & Ebert, D.D. (2021). *Doing Meta-Analysis with R: A Hands-On Guide*. Boca Raton, FL and London: Chapman & Hall/CRC Press. ISBN 978-0-367-61007-4

Figure 1. Why is the green arrow from the brown circle to “high multifunctionality” thicker than the others? I don’t think panel B adds much to the figure as it is subsumed by information

Response: This arrow shows that when two or more interventions have a positive effect (additively or synergistically) on the same ecosystem service, they will have a larger impact than single interventions, as indicated by the thicker arrows.

Panel A provides the rationale for why an increasing number of restoration treatments would enhance ecosystem service multifunctionality, while Panel B offers a clear visualization of our hypothesis. Following your comment, we have improved Panel B by adding information on how an increasing number of restoration interventions enhances the evenness of ecosystem service (page 6).

Line – 139 – sentence could be deleted and just start with “as expected, no single”

Response: We agree that this sentence does not provide any results, however, as such guiding lines are useful in the 'methods last' paper format used by *Nature Communications*, we would like to retain this sentence.

Lines 162-163. It says that the identity of the intervention was significant in the model of ecosystem multifunctionality, but I didn't see it clearly stated which intervention or two resulted in the highest multifunctionality. Likewise, this result is discussed generally but not with respect to specific interventions on Lines 257-258.

Response: The statistical model shown here (Table S2) indicates whether including the identity of interventions would improve the model fit. The combination of interventions that resulted in the highest functionality depends on how we calculated multifunctionality. To clarify this, we have now included the following sentences as follows:

(1) *In the Results section: "multifunctionality generally peaked when all four restoration interventions were used (Fig. 3, 4 and S3). However, it should be noted that we observed the highest value of nature conservation-prioritized multifunctionality, which assumes the prioritization of plant diversity and pollination services, when two specific interventions, the addition of farmyard manure and mixed seeds, were used (Fig. 4 and S3)."* (page 8, lines 174-179)

(2) *In the discussion section: "For example, the addition of inorganic fertiliser and farmyard manure synergistically increased above-ground biomass."* (page 13, line 259-260).

"For example, the combined use of soil disturbance and seeding can additively or synergistically interact to enhance specific ecosystem service indicators (e.g., native species richness and biomass)." (page 13, lines 272-274)

"We also emphasise that the best combination of specific intervention strategies will depend on ecological context, such as the type of grassland" (page 14, lines 277-278)

On line 170 the authors say that their multifunctionality indices reflect different stakeholder priorities. I assumed based on that sentence that they had stakeholders what their priorities were. In fact, the authors subjectively placed the different functions into categories. So, the reference to stakeholders is misleading.

Response: The literature referenced here introduces a method that assigns different weights to various ecosystem services based on different assumed priority scenarios, Although the weights we used are somewhat subjective, they can potentially reflect how different stakeholders evaluate ecosystem service multifunctionality, based on our experience and knowledge of the region. Our calculation of multifunctionality is based on two key principles: first, different stakeholders prioritize different ecosystem services; second, all stakeholders require multiple ecosystem services. In our

calculation, the prioritized ecosystem service group was given a 50% weighting, while the other services that support the prioritized group collectively received the remaining 50%. Following your comment, we have reworded this explanation to avoid potential misunderstandings.

- (1) In the results section, we reworded “multifunctionality indices reflect different stakeholder priorities” to “*multifunctionality indices that reflect different scenarios of stakeholder priorities showed a consistently positive relationship with the number of interventions (Fig. 4 and S1). However, it should be noted that we observed the highest value of nature conservation-prioritized multifunctionality, which assumes the prioritization of plant diversity and pollination services” (page 8, lines 171-178)*
- (2) In Figure 4, we add following sentence to guide the readers to understand how we calculated the multifunctionality indices. “*These ecosystem service multifunctionality indices assigned high proportional weights to forage production, plant biodiversity and pollination, carbon stocks and sequestration, and aesthetic value, respectively, as shown on the y axis of panels a-d.*” (page 10)
- (3) In the Methods section, we reword the following sentence to clearly state how we designed these indices: “*These multifunctionality indices weighted the ecosystem services according to scenarios of weighting representing four possible stakeholder objectives: regenerative agriculture, nature conservation, climate mitigation, and aesthetic value, which assigned high weights to forage production, plant biodiversity and pollination, carbon stocks and sequestration, and aesthetic value, respectively. These scenarios were designed to reflect the likely management goals and ecosystem service priorities of several major stakeholder groups in the study region.*” (page 21, lines 484-491)
- (4) In the discussion section, we added following sentence to point out the limitations of our current approach and potential solutions. “*However, determining the exact nature of the best interventions to achieve ecosystem service multifunctionality would require more detailed information on stakeholder priorities, for example, as assessed in social surveys*⁵⁴, and the combination of interventions that best meets these for whole stakeholder communities^{5,55}.” (page 14, lines 292-296)

Figure 5 – panel B. Aren't the calculation for “ecosystem service functionality” and “evenness of ecosystem services” based on the same numbers? It seems like these two measures aren't independent of one another, though admittedly, I wasn't entirely clear how each of these calculations was done.

Response: The calculations for ecosystem service multifunctionality (equally averaged) and evenness are based on the standardized scores of ecosystem service indicators, but they measure different aspects. Multifunctionality reflects the average performance of all ecosystem services, considering them equally. Accordingly, moderately high multifunctionality can be achieved by just a few very high services, or many at moderately high levels. In contrast, the evenness index provides

complementary information as it measures how evenly the performance of these services is distributed, and uniformly low function levels would still be even (Eisenhauer *et al.*, 2018). The two metrics are thus not intrinsically highly correlated.

Reference

Eisenhauer, N., Hines, J., Isbell, F., Van Der Plas, F., Hobbie, S. E., Kazanski, C. E., ... & Reich, P. B. (2018). Plant diversity maintains multiple soil functions in future environments. *Elife*, 7, e41228."

Lines 276 – This paragraph points out some limitations to the study but doesn't note the replication or scaling issues discussed above.

Response: We now added text on the limitations (page 14, lines 305-308) with respect to replication and scaling, as detailed in our response above.

"Given that ecosystem restoration is occurring at far larger scales than those investigated here, future studies should also explore how combinations of different restoration methods impact ecosystem service multifunctionality across diverse locations and at large spatial scales, to inform the design of multiple targeted interventions at the landscape scale^{5,55}."(page 14, lines 303-307)

Lines 284-289 – This is a long, jargon-filled sentence that says there is a need for different targeted interventions to inform multiple targeted interventions, which doesn't need to be stated.

Response: We agree and have now deleted this sentence.

More generally, there is a fair amount of repetition of generalities in the discussion. The text could be tightened up.

Response: Thanks for your suggestion, we have now revised the discussion to reduce the repetitiveness.

Line 400. It sounds like pollination services were measured observing individual plots of 2 x 2 m for 10 minutes in one month. This seems like a highly insufficient sample size to generalize about pollination services. And as noted previously, I question whether pollination services (at least as applied to restoration) can be assessed meaningfully in such a small plot as the measurements in one plot are necessarily going to be influenced by adjacent plots.

Response: The observation time was chosen to coincide with the peak pollination period, based on the following standards: 1. peak flowering time (June); 2. grazing animals excluded from plots; 3. ideal weather for pollinators (i.e., warm, sunny, without strong winds). We agree that multiple measurements across different years and large

scale would better quantify pollination services. We have now acknowledged this limitation in the discussion section (page 15, line 314-318).

“Current methodology for measuring pollination services could also be improved by taking measurements at multiple time points and larger spatial scales to enhance representativeness.”

Line 447 – Should be “species”. Specie is gold coin.

Response: We have corrected the sentence as suggested

Line 497 – “split” is misspelled.

Response: We have corrected the sentence as suggested

Figure S3. The tick label size needs to be increased to be legible.

Response: We have increased the label size of this figure.

Reviewer #2 (Remarks to the Author):

The authors present data on how multiple management interventions affect multiple ecosystem services in grassland ecosystems. This is an important and timely topic, and the authors did a good job of measuring and interpreting important services. I thought the measurements of pollinator visitation and aesthetic measures of flower abundance and diversity were especially novel. The analysis was appropriate for the objectives of the study. The work is explained in detail and could be reproduced by others. I did have a few suggestions on revising the paper.

Response: Thank you for your positive comments and for highlighting the novelty of our work. Please see our response to your comments below.

Line 381. Are there any carbonates in your soil and how were they dealt with? What was the pH of the soil? Carbonates can be large components of some grassland soils and they have to be analyzed as well as SOM.

Response: Good point. We agree that carbonates are an unneglectable carbon pool in grasslands, especially in arid and semi-arid environments with alkaline soils (Liu *et al.*, 2020, Huang *et al.*, 2024). However, in our study ecosystem, the soils were acidic with a pH of 5.43 ± 0.36 (mean \pm standard deviation), and no carbonate was detected in our soil samples. Generally, carbonate levels in UK soils are very low (Manning *et al.*, 2015), especially in soils with low pH. We have added the following sentence in the Methods and Materials section to reflect this (page 18, lines 406-407).

“Given that there was no detectable inorganic carbon in the soil samples, total carbon is taken to equal organic carbon”

References

Huang, Y., Song, X., Wang, Y. P., Canadell, J. G., Luo, Y., Ciais, P., ... & Zhang, G. L. (2024). Size, distribution, and vulnerability of the global soil inorganic carbon. *Science*, 384(6692), 233-239.

Liu, S., Zhou, L., Li, H., Zhao, X., Yang, Y., Zhu, Y., ... & Fang, J. (2020). Shrub encroachment decreases soil inorganic carbon stocks in Mongolian grasslands. *Journal of Ecology*, 108(2), 678-686.

Manning, P., de Vries, F. T., Tallowin, J. R., Smith, R., Mortimer, S. R., Pilgrim, E. S., ... & Bardgett, R. D. (2015). Simple measures of climate, soil properties and plant traits predict national-scale grassland soil carbon stocks. *Journal of Applied Ecology*, 52(5), 1188-1196.

I thought the usage of evenness measures in the ES part of the paper was somewhat novel and interesting. Can you cite some of the evenness literature in this section?

Response: Thank you. We have now cited relevant literature that used similar approach to estimate evenness of multifunctionality in this section (Eisenhauer *et al.*, 2018). Please see page 21, line 482.

Reference

Eisenhauer, N., Hines, J., Isbell, F., Van Der Plas, F., Hobbie, S. E., Kazanski, C. E., ... & Reich, P. B. (2018). Plant diversity maintains multiple soil functions in future environments. *Elife*, 7, e41228."

I agree with the authors that we need more consideration of grazing intensity in these kinds of studies. It was nice to see that they acknowledged that in the discussion.

Response: We are pleased that the reviewer agrees that grazing management (e.g., grazing intensity, grazing animal diversity) is a critical factor controlling ecosystem service provision in grasslands.

Reviewer #3 (Remarks to the Author):

We are glad to have had the opportunity to review the manuscript "Multiple targeted grassland restoration interventions enhance ecosystem service multifunctionality" by Liu et al. This work covers an essential and critical topic, especially in light of global climate change, biodiversity crises, and the ongoing demand for increased agricultural production while conserving nature.

The manuscript focuses on the restoration of agriculturally improved grasslands, examining how different restoration interventions impact ecosystem service provision and ecosystem multifunctionality. The novel and valuable aspect of the study arises from the thorough sampling and from the opportunity to observe the long-term impacts

of restoration interventions using multiple treatments. The comprehensive data gathering and analysis cover a wide range of ecosystem services and their combined multifunctionality, offering an integrated and thorough knowledge of the ecosystem improvements achieved.

The fact that the study focuses on grasslands is also relevant - grasslands are often overlooked in conservation and restoration efforts. Despite their critical role in ecosystem service provision and biodiversity, grasslands are experiencing significant degradation worldwide, including in Europe, as also clearly pointed out by the authors.

The manuscript is interesting, well-composed, covers an important and timely topic, and has high applicable value for real-life restoration actions. The methodology and analysis are sound and well explained, meeting the quality requirements and standards of the field. However, few aspects can be further elaborated or clarified.

Response: Thank you for your positive comments. We appreciate your highlighting of the high application value of our manuscript for informing real-life restoration actions.

General comments:

Comment 3.2 The study focuses on improved agricultural grasslands, while the introduction and discussion cover grasslands in a very general sense. It should be clear from the introduction to the discussion that the study considers (and obtained results apply to) only one type of grasslands - species poor agriculturally improved grasslands. It is important, as grasslands sensu lato include a variety of ecosystems, ranging from intensively managed agricultural grasslands to semi-natural and natural habitats with native, characteristic, self-assembled plant compositions. The range of potential services they CAN provide varies greatly (e.g. Lindborg et al. 2023), irrespective of the stakeholder interests. Thus, the restoration goals also vary, together with the set of suitable restoration interventions. For example (as also shown in this study), fertilization and seed addition can decrease species richness and homogenize species composition. In ancient (never plowed, never seeded, or fertilized) semi-natural grasslands that hold very high conservation value (e.g. Heinsoo et al. 2020), it would be almost a crime to promote prioritization of biomass, vegetation carbon stock, or even perhaps overall ecosystem multifunctionality if it threatens those conservation values. Thus, to avoid inadvertently promoting practices that could be harmful in some cases, please add a short paragraph acknowledging that there are multiple grassland types, each with different values and uses, and each requiring its own set of restoration interventions. It must be clear that careful consideration of approaches is needed depending on the target ecosystem.

Response: We thank the reviewer for their insightful suggestion. We totally agree with the point that the best approaches to restore grassland will depend on the condition and type of target ecosystem and/or stakeholder priorities, and our experiment is

based on one type of grassland (i.e. 'improved' agricultural grassland), in which we test the potential of using multiple restoration interventions to enhance ecosystem service multifunctionality. We also acknowledge that different grassland ecosystems, ranging from intensively managed to semi-natural habitats, offer differing ecological services and have distinct management priorities. Thus, restoration strategies must be tailored to the specific ecological context of each grassland type to avoid inadvertently promoting practices that could undermine conservation efforts in more sensitive ecosystems.

Following your suggestion, we now elaborate on this point by adding a paragraph to the revised manuscript as follows:

"We also emphasise that the best combination of specific intervention strategies will depend on ecological context, such as the type of grassland. It is essential to recognize that grasslands are not a homogenous ecosystem type; they range from species-rich, semi-natural habitats with high conservation value to intensively managed, agriculturally improved grasslands that meet the demand for food production. These differences have profound implications for the ecosystem services they provide and the most appropriate management and restoration interventions required. For instance, while fertiliser addition may enhance productivity in agriculturally improved grasslands, such practices should be avoided when restoring grasslands for biodiversity. Therefore, restoration approaches must be carefully considered and tailored to the specific ecological context and restoration objectives of each grassland type." (page 14, lines 277-287)

It might also be that improved service provision or increased multifunctionality at the observed time point does not always mean a 'better' or more resilient system in the long run, which should also be addressed. See also DeCock et al., 2023.

Response: We agree that the stability and sustainability of an ecosystem does not necessarily increase with multifunctionality, as measured here. Long-term measurement of ecosystem service indicators is needed to address this issue in the future. In the revised manuscript, we now include the following sentence:

"from a temporal perspective, the stability and sustainability of ecosystem service multifunctionality should also be explored in future studies via long-term observations⁶¹." (page 15, lines 317-319):

The discussion could be revised to reduce some repetitiveness of the findings and add a more thorough synthesis with previously published research on ecosystem multifunctionality. If we may recommend, our own recent study had quite a similar approach but showed that both multitrophic diversity and ecosystem service multifunctionality could be simultaneously increased with grassland restoration (Prangel et al. 2024).

Response: Thank you for your suggestion regarding the literature, which is very relevant. We thoroughly revised the discussion section to reduce repetitiveness. Additionally, we incorporated a discussion of previous literature on ecosystem multifunctionality, following your recommendations, including the mentioned article

“Moreover, the sustainable delivery of ecosystem services multifunctionality may require the restoration of multitrophic biodiversity, rather than plant diversity alone^{49,50}.”(page 12, lines 237-239)

“This result aligns with common practice in restoration to combine two different approaches (e.g., fertiliser and seeding) to restore nutrient-poor degraded grasslands⁵¹.” (page 12, lines 243-244)

“However, determining the exact nature of the best interventions to achieve ecosystem service multifunctionality would require more detailed information on stakeholder priorities, e.g. as assessed in social surveys⁵⁴, and the combination of interventions that best meets these for whole stakeholder communities^{5,55}.” (page 14, line 292-296)

Very small details that might be helpful to authors:

Figure 3 (b) is quite hard to read; it is not easy to understand which group is in which figure (especially as the legend and figures are not in the same order, at least in the end - soil related vs nitrogen-phosphorus seems to be switched). Consider changing the legend or adding subfigure titles.

Response: Thank you for pointing this out. We’ve modified the figure to make sure the legend and figure are in the same order (page 9).

Figure 3. The relationship between number of interventions and ecosystem service multifunctionality (a) and individual ecosystem service indicators (b). Ecosystem service multifunctionality was calculated by equally weighted averaging. Ecosystem service multifunctionality calculated using the multiple threshold method showed similar trends (Fig. S3).

Lines 45-46: Suggest removing “also” from the sentence and adding “provisioning services” to the list. “Grasslands also provide a wide range of provisioning, regulating, supporting, and cultural ecosystem services such as food production, carbon sequestration...”

Response: Thank you, we now changed this sentence to:

“Grasslands are the largest terrestrial biome on Earth¹ and support the livelihoods of more than 1 billion people². Beyond their provisioning services, grasslands also provide a wide range of regulating, supporting, and cultural ecosystem services such as carbon sequestration, soil fertility maintenance, pollination, and aesthetic value³.” (page2, lines 43-45).

Lines 79-83: This sentence is quite long and contains a lot of information. It could be split into two sentences.

Response: We have adjusted this sentence as suggested to:

“For instance, mixed seed addition can promote plant diversity^{28,29}, with associated benefits for pollinators and cultural ecosystem services^{12,30}. Low levels of fertiliser and/or manure inputs can replenish depleted soil nutrients and organic matter, thereby

*helping to maintain forage production and increase soil carbon sequestration*31–33 (page 4, lines 78-81).

Line 84: change 'word' to 'world'.

Response: We have corrected the sentence (page 4, line 83).

Lines 87-89: “Alternatively, multiple interventions could lead to neutral or negative effects on ecosystem service multifunctionality if the interventions generally exert divergent or antagonistic impacts on ecosystem services.” Please add an example.

Response: We have added a reference to illustrate this possibility (page 4, line 88).

Reference:

Halassy, M., Singh, A. N., Szabó, R., Szili-Kovács, T., Szitár, K., & Török, K. (2016). The application of a filter-based assembly model to develop best practices for Pannonian sand grassland restoration. *Journal of Applied Ecology*, 53(3), 765-773.

Line 215: Add 'than' to “As such, our results suggest that the use of multiple restoration interventions, rather than single interventions...”

Response: We have corrected the sentence as suggested (page 12, lines 217-218).

Lines 295-296: Add “be” in the sentence. “The wider environmental footprint (e.g., antibiotic resistance) of restoration interventions also needs to be considered.”

Response: We have corrected the sentence as suggested (page 15, line 313).

Line 298: make a correction: “...multifunctionality should also be explored in future studies via long-term observations.”

Response: We have corrected the sentence as suggested (page 15, line 317).

Lines 405-406: make a correction: "The observation quadrat was 2 m × 2 m, with the quadrat placed at the center of the treatment plot."

Response: We have corrected the sentence as suggested (page 19, line 422).

Line 440: “flower” should be in plural. “The aesthetic value was indicated by the abundance and diversity of flowers.”

Response: We have corrected the sentence as suggested (page 20, line 461).

Line 447: make a correction: “...flowering plant species.”

Response: We have corrected the sentence as suggested (page 20, line 468).

Line 459: make a correction: “we also calculated the evenness of ecosystem service indicators using the ‘vegan’ package...”

Response: We have corrected the sentence as suggested (page 22, line 480-481).

Line 530: make a correction: "Moreover, simple linear regression..."

Response: We have corrected the sentence as suggested (page 23, line 552).

References

DeCock, E., Moeneclaey, I., Schelfhout, S., Vanhellemont, M., De Schrijver, A., Baeten, L., 2023. Ecosystem multifunctionality lowers as grasslands under restoration approach their target habitat type. *Restor. Ecol.* 31, 1–10.

Heinsoo, K., Sammul, M., Kukk, T., Kull, T., & Melts, I. (2020). The long-term recovery of a moderately fertilised semi-natural grassland. *Agriculture, Ecosystems & Environment*, 289, 106744.

Lindborg, R., Hartel, T., Helm, A., Prangel, E., Reitalu, T. and Ripoll-Bosch, R., 2023. Ecosystem services provided by semi-natural and intensified grasslands: Synergies, trade-offs and linkages to plant traits and functional richness. *Applied Vegetation Science*, 26(2), p.e12729.

Prangel, E., Reitalu, T., Neuenkamp, L., Kasari-Toussaint, L., Karise, R., Tiitsaar, A., Soon, V., Kupper, T., Meriste, M., Ingerpuu, N. and Helm, A., 2024. Restoration of semi-natural grasslands boosts biodiversity and re-creates hotspots for ecosystem services. *Agriculture, Ecosystems & Environment*, 374, p.109139.

*This review was compiled jointly by an early-career researcher and the advisor.

Elisabeth Prangel, Aveliina Helm (University of Tartu, Estonia)

Reviewer #4 (Remarks to the Author):

"I co-reviewed this manuscript with one of the reviewers who provided the listed reports. This is part of the Nature Communications initiative to facilitate training in peer review and to provide appropriate recognition for Early Career Researchers who co-review manuscripts."

Response: Thank you for reviewing our manuscript and your constructive comments! Please see the responses above.

Responses to reviewer comments in blue

REVIEWERS' COMMENTS

Reviewer #1 (Remarks to the Author):

I appreciate that the authors have addressed many of my and other reviewers' comments by rewording portions of the text which has improved the paper. I particularly recognize the effort they made by adding the power analyses.

Response: Thank you for your kind comments and for acknowledging our previous revisions.

However, a few of my initial concerns remain. First, the authors still make multiple broad sweeping statements that are made about restoration interventions based on an experiment in which treatments were conducted in 3 x 3 m plots replicated 3 times each at a single site. I list a few examples of their overstatements below...

Line 83, 101-102 – the authors say that multiple interventions have not been tested in “real world restoration trials”. Their 3 x 3 m plots are not “real world restoration trials” though they imply that in lines 101-102. They are reporting on small-scale restoration experiments. As I mentioned in the last review, they over-sell the scale and real-world practicality of their results. Also, the text is nearly identical in two adjacent paragraphs, so it is redundant. I strongly recommend deleting the sentence in line 100-102.

Response: We have removed the term “real-world” from the sentence in Line 83 and deleted the sentence in Lines 100-102, as requested.

Line 220 says that the authors “Overall, our findings identify a novel strategy for the restoration of grasslands that could potentially be extended to other managed ecosystems to aid simultaneous deliver of multiple ecosystem services.” Again, this is inaccurate. They don't identify a novel restoration strategy. These are all standard restoration strategies that are sometimes implemented at the same time and sometimes not – e.g. lots of managers seed, including N fixing species, and inorganic fertilizer in grasslands and other terrestrial ecosystems. It is seen that combination used in temperate grasslands and forests by restoration practitioners across multiple hectares and not just in experimental plots. Most practitioners don't use both inorganic fertilizer and manure because of the costs of multiple interventions.

Response: We agree that all the treatments included in the study are standard restoration approaches, and "novel" was meant to highlight the combined application of these standard methods. We have reworded this sentence to prevent any potential misinterpretation.

“Overall, our findings identify a combination strategy for the restoration of grasslands that could potentially be extended to other managed ecosystems to aid simultaneous deliver of multiple ecosystem services.” (lines 177-180)

Line 262 says “While the number of restoration interventions can be a strong predictor of ecosystem service multifunctionality”. Figure 3b does not suggest that this relationship is “strong”. Yes – some of the lines increase with the number of interventions, but many of them are flat and a couple go down. I think it should just “can be a predictor.”

Response: We have revised this sentence as suggested

“While the number of restoration interventions can be a predictor of ecosystem service multifunctionality, it is important to note that the identity of the intervention selected will also determine the outcomes of grassland management.” (line 219-221)

Figure 1b. I still don’t think panel B adds much to the figure as the same information is in panel A in a different format. It seemed clear from figure A and the multiple times the authors state it in the text that they think the number of interventions will increase ecosystem multifunctionality.

Response: We have removed panel B from Figure 1 as suggested.

On line 170 the authors say that their multifunctionality indices reflect different scenarios stakeholder priorities. In the figure they call them “different management objectives”. I think the latter word is more accurate and should be used throughout in the text which would simplify the text considerably.

Response: We have updated the text to use “different management objectives” throughout, as suggested.

Lines 287-295. I appreciate that the authors have now explained that stakeholders surveys would be necessary to assess stakeholder priorities. However, I think the paragraph could be shortened considerably by just saying that “The selection of restoration interventions should be guided by management priorities.” The rest of the text is fairly long-winded without saying much. I think that throughout (e.g. line 486) the authors should switch from stakeholder priorities to management objectives.

Response: We have shortened this paragraph as suggested and switched “stakeholder priorities” to “management objectives” in line 486, as suggested.

In the closing paragraph the authors only passingly consider the costs of implementing multiple restoration interventions, but costs are a key consideration for land managers. Yes these methods are commonly used but managers tend to pick and choose the ones that they find to be most effective to achieve their goals given cost limitations.

Response: We completely agree that cost is a key consideration for land managers, especially in resource-constrained regions. We have now revised these sentences to reflect this.

“Determining the exact nature of the best interventions to achieve ecosystem service multifunctionality would require more detailed information on stakeholder priorities, for example as assessed in social surveys, as well as the financial and labour costs of implementing these interventions” (lines 246-249)

“We acknowledge that an increasing number of interventions may inevitably require more financial investment for restoration, which may limit the scalability in resource-constrained regions. Nevertheless, the interventions used here are all commonly used in grassland management, albeit usually singularly, and do not involve large additional investment in resources or machinery. A growing number of environmental land management schemes offer financial support for farmers to restore degraded grasslands, and these could be effectively employed to fund grassland restoration schemes using multiple interventions, as proposed here” (lines 276-298)

More generally, there is a fair amount of repetition of generalities in the discussion. The text could be tightened up, despite the fact that the authors say they already did that.

Response: We have shortened the discussion based on your comments above.

Pollinator measurements. I still contend that observing individual plots of 2 x 2 m for 10 minutes in one month is insufficient to characterize pollinators. I know the authors measured many variables but that doesn't mean that how they measure different variables should be of low rigor.

Response: We would like to draw your attention to the following sentence in the discussion, which acknowledges this potential limitation:

“Current methodology for measuring pollination services would also be improved by taking measurements at multiple time points and larger spatial scales to enhance representativeness.” (lines 268-270)

Reviewer #2 (Remarks to the Author):

I thought that the authors did a great job in responding to the reviewers and the revision is greatly improved.

Response: Thank you again for reviewing this manuscript!